https://doi.org/10.1038/s41467-022-31987-w · **OPEN**

# Bioinspired asymmetric amphiphilic surface for triboelectric enhanced efficient water harvesting

Song Zhang [1], Mingchao Chi[1], Jilong Mo[1], Tao Liu [1], Yanhua Liu[1], Qiu Fu[1], Jinlong Wang [1], Bin Luo[1], Ying Qin [1], Shuangfei Wang[1] & Shuangxi Nie [1✉]

The effective acquisition of clean water from atmospheric water offers a potential sustainable solution for increasing global water and energy shortages. In this study, an asymmetric amphiphilic surface incorporating self-driven triboelectric adsorption was developed to obtain clean water from the atmosphere. Inspired by cactus spines and beetle elytra, the asymmetric amphiphilic surface was constructed by synthesizing amphiphilic cellulose ester coatings followed by coating on laser-engraved spines of fluorinated ethylene propylene. Notably, the spontaneous interfacial triboelectric charge between the droplet and the collector was exploited for electrostatic adsorption. Additionally, the droplet triboelectric nanogenerator converts the mechanical energy generated by droplets falling into electrical energy through the volume effect, achieving an excellent output performance, and further enhancing the electrostatic adsorption by means of external charges, which achieved a water harvesting efficiency of 93.18 kg/m$^2$ h. This strategy provides insights for the design of water harvesting system.

[1] School of Light Industry and Food Engineering, Guangxi University, Nanning 530004, PR China. ✉email: nieshuangxi@gxu.edu.cn

With the worldwide growth in population and energy consumption, increasingly severe water scarcity presents challenges to the scientific and engineering community[1–3]. Atmospheric water is a ubiquitous environmental resource with an existing total quantity of approximately $1.29 \times 10^{16}$ kg, which is equivalent to three times the world's total annual water use[4–6]. Fog collection, as a typical capture scheme for atmospheric water, enables rapid on-site production[7–9]. Much of the collection methods are inspired by biological traits that adaptively evolved to allow primitive organisms to cope with harsh environments, such as cactus spines (surface energy gradients and asymmetric curvature), elytra of Namib Desert beetles (surface chemistry), and an inner wall of pitcher plant (slippery surface)[10–12]. Therefore, ingeniously imitating these interesting biological features is an effective way to alleviate water scarcity[9,13,14].

Artificial surfaces that mimic a single organism may be limited in one way or another in fog harvesting[15,16]. For example, the water affinity of asymmetric spine surfaces would lead to fog shielding; regionalized droplets on superhydrophilic/hydrophobic surfaces cannot be shed until larger than the critical size; and hydrophobic lubricants on smooth surfaces would hinder fog capture. Thus, it is crucial to ingeniously combine multiple biomimetic principles for designing synergistic mechanism of droplet nucleation and removal to simplify large-scale preparation[13,14]. In addition, electron transfer has been demonstrated in liquid-solid interface[17–19]. Theoretically, fog or water droplets on the collector surface would also be affected by the charge interaction. However, triboelectrically induced interfacial interactions during water harvesting have been rarely investigated.

In this work, an asymmetric amphiphilic surface combining notable features of cactus spines and elytra of Namib Desert beetles is designed for triboelectric-enhanced water harvesting. Amphiphilic cellulose ester coating (ACEC) provides hydrophilic sites for the efficient nucleation of droplets, its hydrophobic component and the asymmetric spine enhance droplet removal after coalescence; hydrophobic channels with a smooth interface contribute to the refreshment of surface droplets. Afterward, the collected droplets are used as a charge source to trigger droplet triboelectric nanogenerators (D-TENG) for enhancing water harvesting. More importantly, electrostatic adsorption induced by triboelectrification between droplets and collectors is investigated. And the adsorption effect could be further enhanced by the external charge, achieving high efficiency for water harvesting.

## Results

**System design and working principle**. Water-harvesting devices inspired by the cactus spines and elytra of Namib Desert beetles involve structural design and surface chemistry (Fig. 1a). Biomimetic cactus arrays were fabricated using laser-engraved fluorinated ethylene propylene (FEP) membranes for Laplace driving forces. ACEC was synthesized by nucleophilic substitution for surface chemistry, in which the hydrophilic and hydrophobic components coexist. Hydroxyl groups on cellulose act as hydrophilic sites for droplet nucleation, and grafted 10-undecenoyl groups provide hydrophobic sites to facilitate the removal of water droplets (Fig. 1b).

The fog was captured and condensed into tiny water droplets and transported to the hydrophobic channel. Under gravity and hydrophobic repulsion, the water droplets slip along the smooth hydrophobic channel. Afterward, the surface of FEP was charged due to liquid-solid contact, showing a negative potential, which in turn produces electrostatic adsorption to the fog. To reinforce this phenomenon, a D-TENG was fabricated with narrow copper foil

as the upper electrode, FEP as the dielectric layer, and conductive cloth tape (CCT) as the lower electrode. Through rectification, the electrical energy generated by the droplets hitting enhanced electrostatic adsorption and further improved the water collection effect (Supplementary Note. 1 and Supplementary Fig. 1).

**Properties of ACEC**. There are three hydroxyl groups on each glucose unit of cellulose, making it naturally hygroscopic and causing water to adhere[20,21]. To weaken this adverse effect, an amphiphilic cellulose ester coating was prepared by grafting hydrophobic 10-undecenoyl chloride into cellulose (Fig. 2a). The chemical properties of ACEC were investigated by elemental analysis, solid-state $^{13}C$ nuclear magnetic resonance (NMR) spectroscopy, Fourier transform infrared spectroscopy (FTIR), and X-ray electron spectrometry (XPS). As shown in Fig. 2c, the signals at ~138.6 and 114.5 ppm, 106.6–60.8 ppm, and 36.2–22.1 ppm belong to the carbon atoms of the 10-undecenyl terminal olefin, the cellulose backbone, and the saturated aliphatic chain, respectively[22,23]. Compared with the XPS spectrum of microcrystalline cellulose (MCC), the C/O element ratio of ACEC increased from 57.66 to 73.34% (Supplementary Fig. 2). Furthermore, the C1s spectra of MCC and ACEC were analyzed. Three peaks presented at 287.6, 285.9, and 284.3 eV for the structures of O-C-O, C-O, and C-C/H, respectively (Fig. 2d)[24,25]. A peak at 288.7 eV appeared in the ACEC, belonging to the O-C=O (Fig. 2e)[26]. In addition, the peak at 3334 $cm^{-1}$ (hydroxyl) decreased, and peaks appeared at 3076 (C=C), 1742, and 1643 (C=O) $cm^{-1}$ in the FTIR spectrum (Supplementary Fig. 3)[26]. These results indicated that ACEC was successfully synthesized. The average elemental composition of ACEC was further analyzed, and its degree of substitution was found to be 2.14. In addition, compared with the smooth surface of MCC, ACEC shows a porous structure, which provides more gas-solid interface (Fig. 2f).

SEM was employed to observe the morphology of the FEP spines and ACEC@FEP. The FEP spine was an irregular triangular prism with a smooth upper surface (Fig. 2gi–ii). Dense spherical protrusions appeared on the side, which might be the FEP spherulite formed by laser melting (Fig. 2i). The spine tips of the ACEC@FEP appeared as hetero-shaped cones with a uniform porous coating (less than 5 μm) on the surface (Fig. 2hi–ii). With the gradual increase of the spine width, the spine presented as a prismatic table with the upper and sides coated with ACEC (Fig. 2j, k). Compared with FEP, the ACEC@FEP surface exhibited more C and O and sporadic F elements (Fig. 2l and Supplementary Fig. 4). The formation of these pores might be caused by solvent volatilization, which would provide more nucleation sites for water molecules. In addition, the edges and sides were coated with ACEC (Fig. 2k), which facilitated the capture of droplets on the sides and the transfer of droplets to the upper surface. Furthermore, both FEP and ACEC@FEP exhibited a similar surface free energy, which could serve as a smaller energy barrier when the droplet is transferred from the spines to the hydrophobic channel.

**Wettability and robustness**. Surface wettability plays an important role in droplet nucleation and removal, which can be evaluated by a contact angle measurement. As shown in Fig. 3a, MCC presented a contact angle of 0° due to its natural hygroscopicity. ACEC showed apparent hydrophobicity (93 ± 1°), which stems from the increased density of hydrophobic domains (grafted long aliphatic chains), as well as the porous structure of the coating to achieve a Wenzel state. Afterward, the surface free energy of ACEC and FEP was calculated. The surface free energy ($\gamma_2$) of ACEC was 21.43 mN/m with a polar component ($\gamma_2^p$) of 1.94 mN/m and

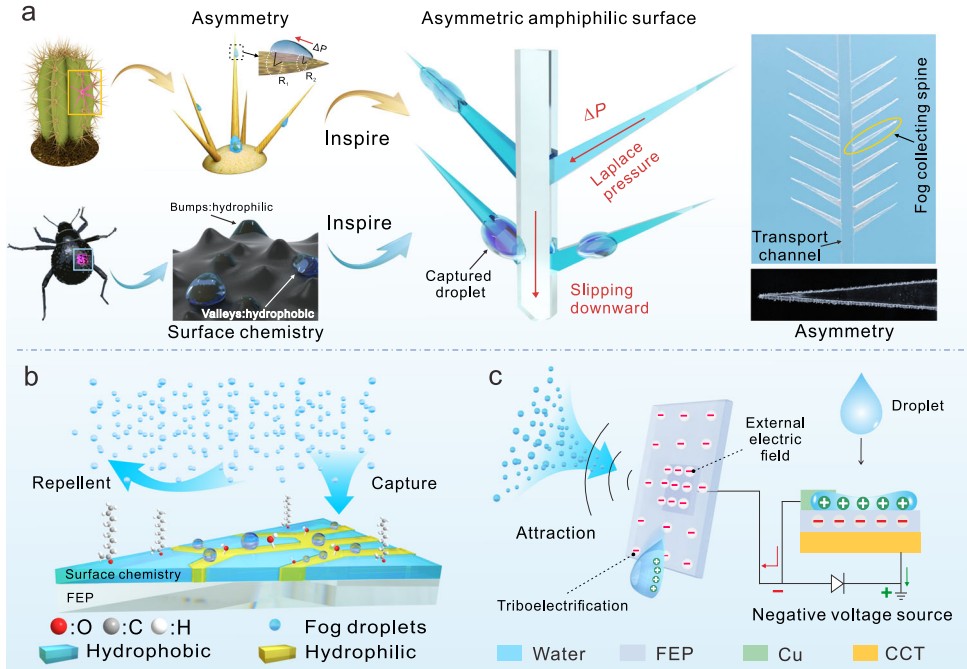

**Fig. 1 Design of the water harvesting system. a** Schematic illustration of asymmetrical smooth surface inspired by the asymmetry of cactus spines and the hydrophilic-hydrophobic surface of the beetle elytra. **b** Schematic diagram of the biomimetic hydrophobic-hydrophilic coating; the fog was repelled in the hydrophobic area and attracted in the hydrophilic area. **c** Electrostatic-assisted enhancement of water adsorption: electrostatic sources including triboelectrification between the droplet and FEP as well as an additional charge from the D-TENG.

dispersion component ($\gamma_2^p$) of 19.49 mN/m (Fig. 3b). The dispersion component was much larger than the polar component owing to the long aliphatic chain (nonpolar) replaces the original hydroxyl (polar) on the cellulose skeleton to a certain extent. Moreover, the value of the surface free energy and corresponding component of ACEC and FEP were similar, which would be conducive to the low interfacial tension ($\gamma_{12} = 1.67$ mN/m) and high adhesion work ($Wa = 33.56$ mN/m) on the two-phase interface based on the formula 1 and 2.

$$\gamma_{12} = \gamma_1 + \gamma_2 - \frac{4\gamma_1^d\gamma_2^d}{\gamma_1^d + \gamma_2^d} - \frac{4\gamma_1^p\gamma_2^p}{\gamma_1^p + \gamma_2^p} \quad (1)$$

$$Wa = \gamma_1 + \gamma_2 - \gamma_{12} \quad (2)$$

Contact angle hysteresis (CAH) on the FEP and ACEC @ FEP surfaces was further investigated (Fig. 3c). The advancing contact angles of the FEP and ACEC@FEP were similar (108° and 111°), while the receding contact angles (61° and 85°) and the CAH (47° and 26°) were quite different. The larger contact angle difference on ACEC@FEP was mainly due to the surface heterogeneity. Compared with FEP, the retraction of droplet contact line on ACEC@FEP needs to overcome the extra barrier of hydrophilic domains, contribute to a smaller receding contact angle, whereas can still realize. Such a surface enables the hydrophilic effect while avoiding the pinning of droplets.

To evaluate the robustness of the coatings, droplet sliding, as well as immersion, were performed to observe the morphology and contact angle changes. As can be seen in Fig. 3d, the surface morphology of ACEC@FEP hardly changed after the water droplets slid down continuously for 10,000 times (Supplementary Fig. 5). The contact angle remained stable and the coating did not peel off after soaking in water for different time (Supplementary Fig. 6). Additionally, the adhesion strength between ACEC and FEP was investigated using the micro-scratch method. As the load

increased, the coating peeled off, the acoustic signal and penetration depth changed abruptly, and the adhesion strength of $1121 \pm 93$ mN was obtained (Fig. 3f). These results demonstrate the feasibility and potential of ACEC@FEP for water harvesting.

**Structural optimization of asymmetric amphiphilic surface.** The detailed process of water harvesting by FEP and ACEC@FEP spines was observed using an industrial microscope. Both materials showed the characteristics of fog harvesting: the fog nucleated at the spikes and lateral edges, gradually condensed into small water droplets, and then progressed toward the root while merging with adjacent small droplets. Compared with FEP, ACEC@FEP exhibited faster fog collection and transport ($t_1 = 13.3$ s) (Supplementary Figs. 7, 8 and Fig. 4a). This phenomenon is attributed to the synergy between the hydrophilic and hydrophobic components of ACEC. In addition, because of the continuous coverage of the ACEC on the upper part and side (Fig. 2j), the small droplets that condensed on the side could also merge with the droplet moving toward the root. According to the Laplace formula (S3), spines with different $\sin\alpha$ were prepared by adjusting the height and width, and their water-harvesting efficiency was investigated (Supplementary Note. 2 and Supplementary Figs. 9, 10). Compared with the symmetric structure, the asymmetric presented a higher water-harvesting efficiency. Especially at a height of 10 mm and a width of 1 resulted in the smallest $\sin\alpha$ value and the best harvesting efficiency.

As expected, even angle between the spine and the hydrophobic channel increased from 30° to 90°, the droplet could cross into the hydrophobic channel (Fig. 4b). The force analysis showed that when the spine is horizontal, the droplet is subjected to the Laplace pressure ($F_L$) toward the root, downward gravity ($G$), and opposite adhesion forces ($F_{V1}$, $F_{V2}$). When the spine is inclined, $G$ can be decomposed into $F_y$ and $F_x$, and $F_x$ is coaxial with $F_L$ (providing an additional driving force for the droplet) and

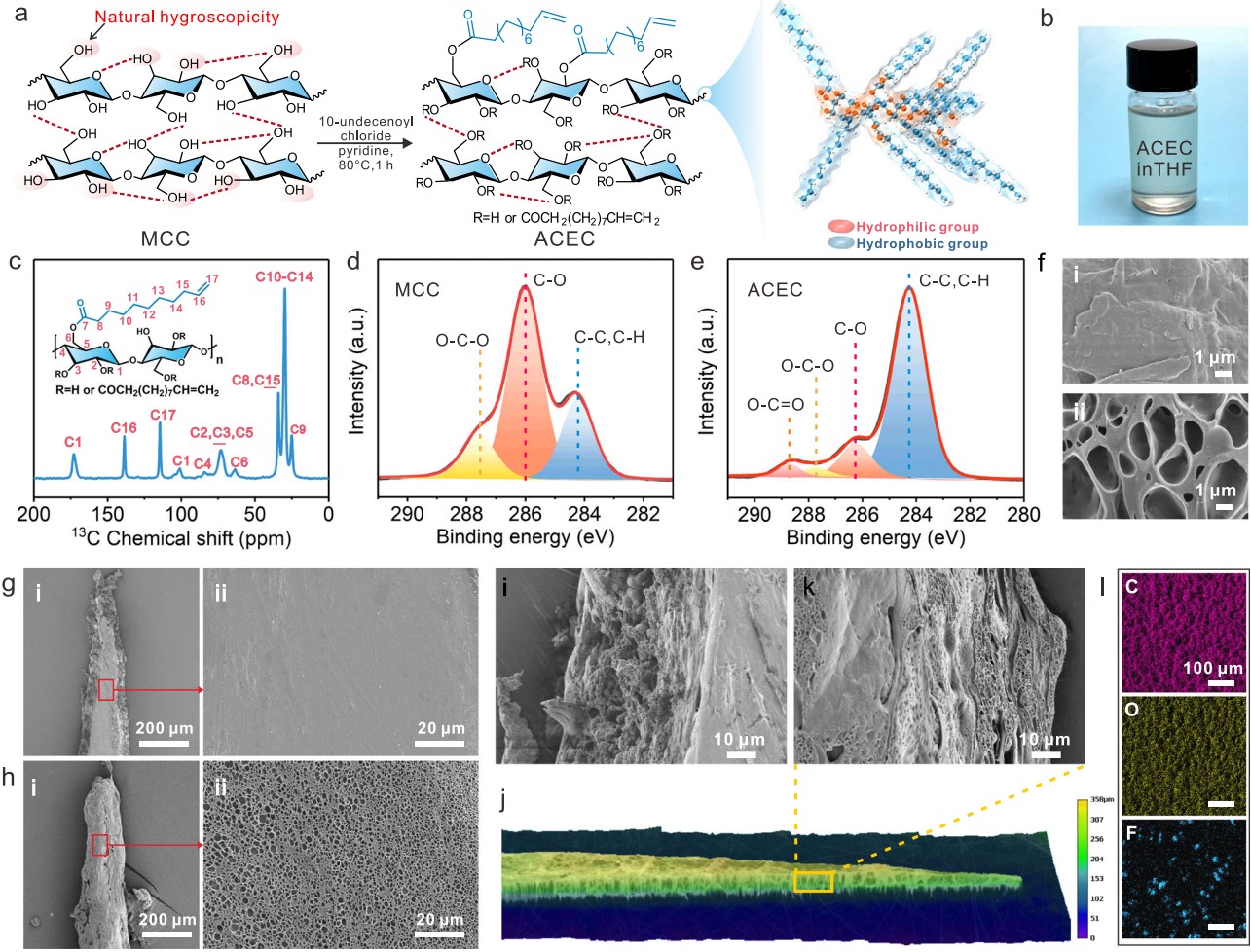

**Fig. 2 Preparation and characterization of ACEC. a, b** Reaction scheme of ACEC and photo of ACEC dissolved in THF. **c** Solid-state ¹³C NMR spectra of ACEC. **d, e** High-resolution C1s XPS spectra of MCC and ACEC. **f** SEM images of MCC and ACEC. **g, h** SEM images of FEP spines and ACEC@FEP spines. **i** SEM image of the side of the FEP spine. **J, k** 3D image of the complete ACEC@FEP spine structure and SEM image of its side. **l** EDX-mapping images of C, O, and F elements of ACEC@FEP.

decreased with an increase in the vertical angle (Fig. 4c), which leads to a decrease in the droplet transport speed. Nevertheless, there is a borderline between the spine and the hydrophobic channel. When the droplet crept into the hydrophobic channel, it would inevitably be dragged due to the hydrophilicity of ACEC, resulting in prolonged shedding time (t₃, Supplementary Figs. 11 and 12). Additionally, since the spines dominate the water harvesting, more spines within the same height (i.e., a higher ratio of spine area) are beneficial for water harvesting (Supplementary Fig. 13). Therefore, a larger additional driving force and shorter borderline, as well as higher spine area ratio together, result in the highest water-harvesting efficiency at 60°. Therefore, an angle of 60° was selected for the following tests.

The smooth surface and physical boundaries of the hydrophobic channel facilitated a rapid slide-off of the droplet. The optimal water-harvesting rate (85.47 kg/m² h) was obtained with a width of 2 mm, and a decrease or increase in width resulted in a delay in the droplet movement due to excessive solid-liquid contact lines or inefficiency/ineffective droplet coalescence (Fig. 4e, Supplementary Note. 3, and Supplementary Figs. 14, 15).

The ipsilateral and contralateral vertical distances of the spine were further adjusted, and the highest water-harvesting rate was obtained when the ipsilateral and contralateral vertical distances were 2 and 1.2 mm, respectively. Observation of the water harvesting revealed that the droplets had undergone nucleation,

collection, and coalescence with contralateral droplets, and were then held at the junction (Supplementary Fig. 16a). By increasing the vertical distance of the contralateral spines in the same group, droplet adhesion was significantly weakened (Supplementary Fig. 16b). Compared with the uniform force of the droplets in the symmetrical structure, the asymmetry led to a stress concentration, which contributed to the rapid shedding of the droplets (Fig. 4h). Therefore, an array with an ipsilateral vertical distance of 2 mm and contralateral distances of 1.2 mm was selected.

**Output performance of D-TENG.** Organofluorine electrets with excellent charge storage capacities have been widely used in the field of solid-liquid triboelectrification[19,27–29]. Compared with PTFE, the molecular chain of FEP is randomly entangled because of the hexafluoropropylene group, and its film shows better flexibility and interfacial bonding with the metal. Therefore, in this study, a commercial FEP film (25 μm) was employed as the dielectric layer with the conductive cloth tape as the lower electrode, and a thin copper foil (~1 mm) was used as the upper electrode. After a simple assembly in sequence, D-TENG was obtained (Fig. 5a), which behaved as a bias capacitor, and the detailed charge transfer mechanism is shown in Fig. 5b.

After being charged by the droplets, the upper and lower electrode charges are opposite to those of the FEP because of electrostatic induction. When the droplet spreads to the upper

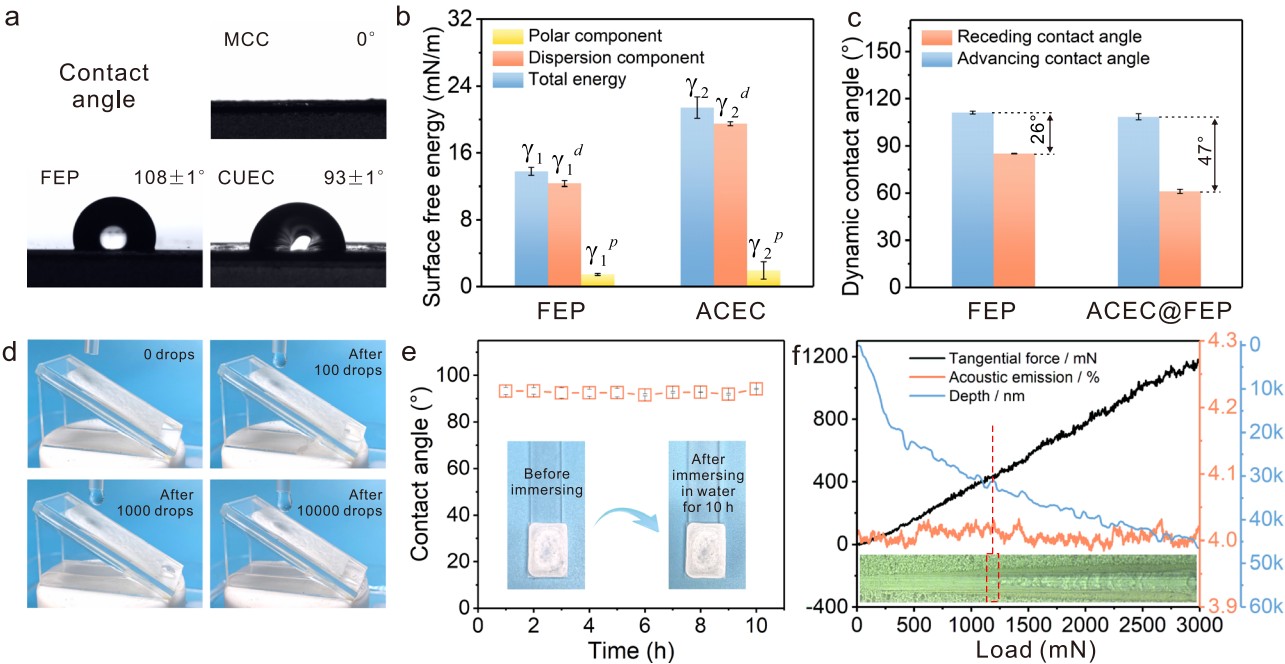

**Fig. 3 Wettability and robustness. a** Photographs of contact angle of MCC, ACEC, and FEP. **b** Surface free energy of FEP and ACEC. **c** Advancing and receding contact angles of FEP and ACEC@FEP. **d** Photographs of ACEC@FEP surface after the water droplets sliding down. **e** Contact angle and photographs after immersion in water for a different time. **f** Adhesion strength between ACEC and FEP.

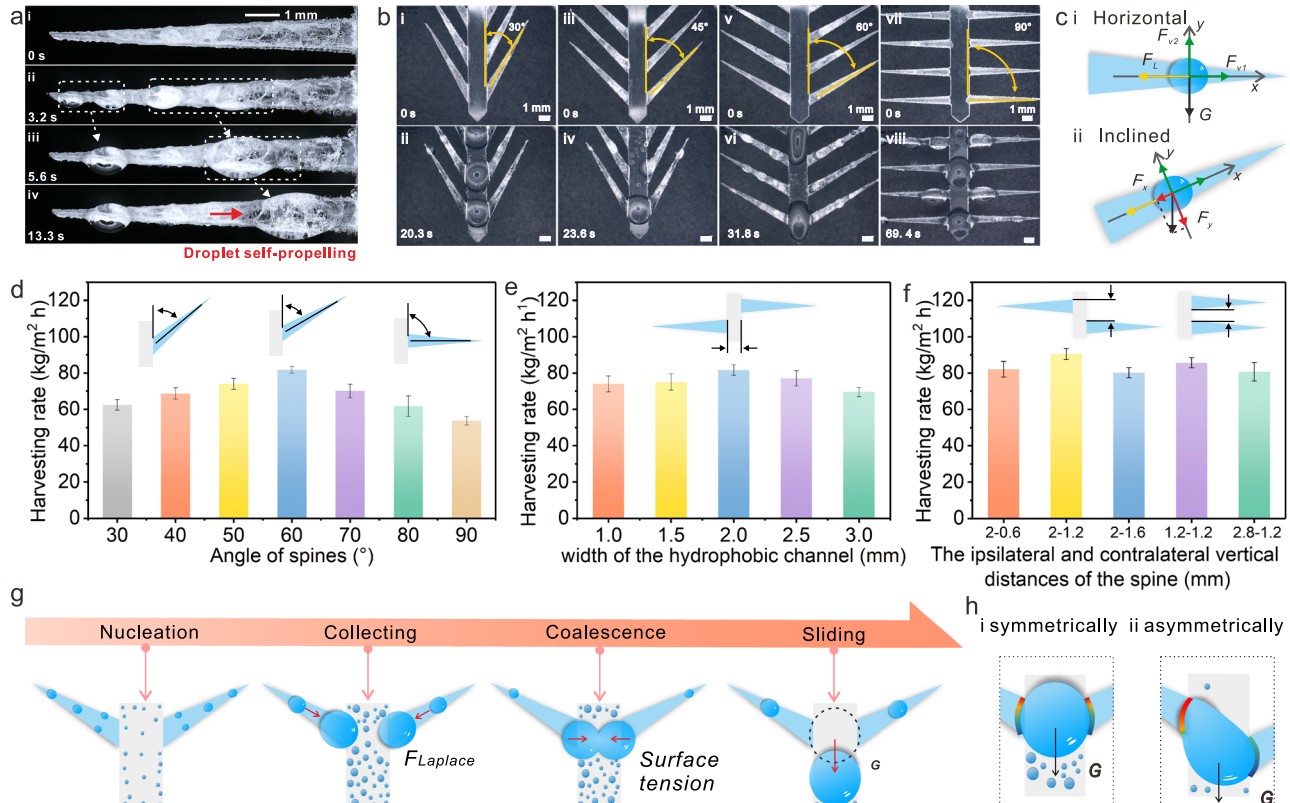

**Fig. 4 Water-harvesting performance of dual bionic structures. a** The harvesting process of the fog on the ACEC@FEP spine. **b** Fog harvesting in an array with different angles between the spine and hydrophobic channel (30, 45, 60, and 90°). **c** Force analysis of the droplets on different spines. **d** Water-harvesting rate of the array with different angles. **e, f** Water-harvesting rate of the array with different hydrophobic channel widths and different vertical distances of the spines; the root width of the spine is 1.2 mm. **g** Schematic diagram of the water harvesting. **h** The force analysis of the droplets on the array with symmetric and asymmetric spines.

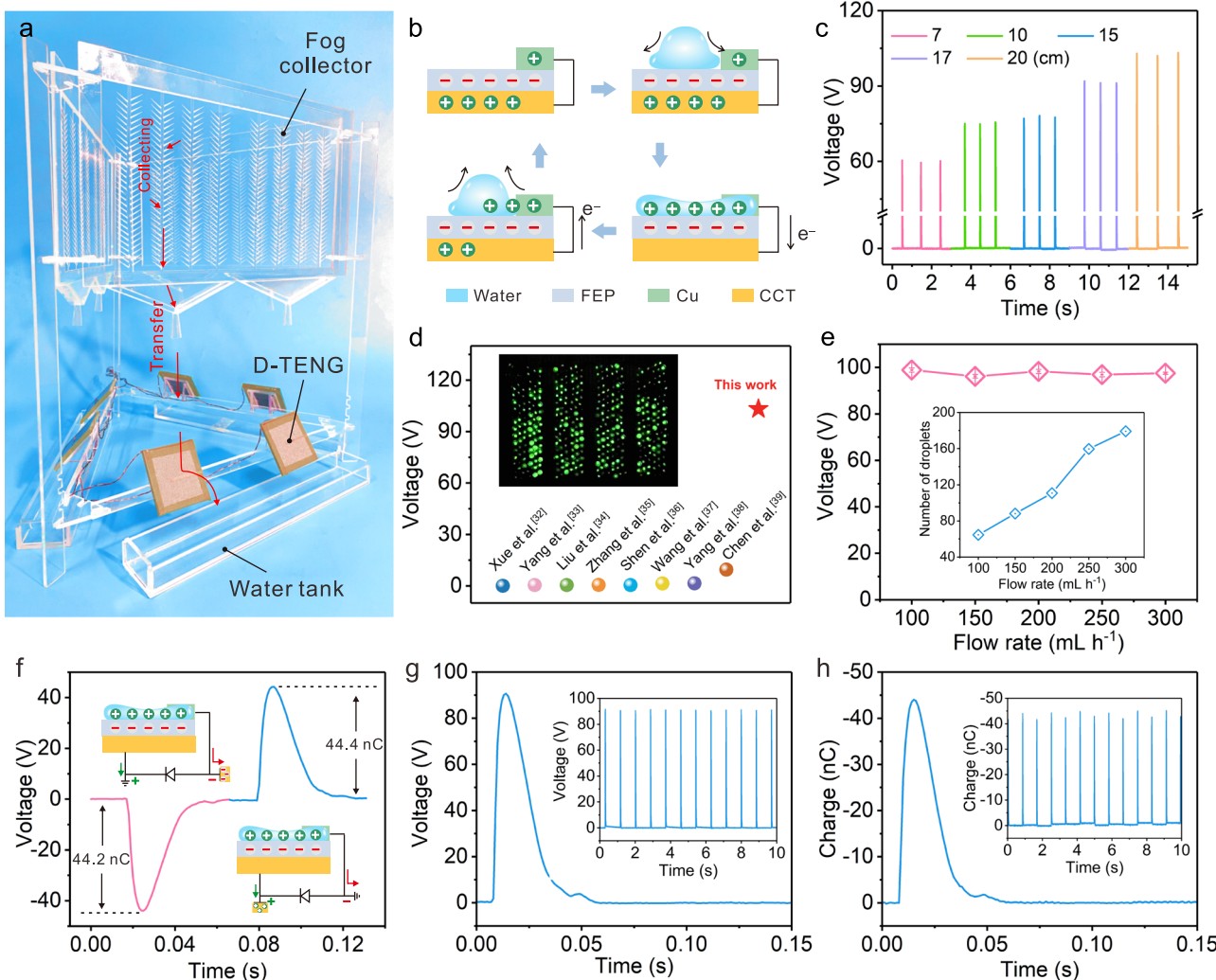

**Fig. 5 Output performance of the D-TENG. a** Self-made platform for fog collection and power generation. **b** Schematic diagram of the working mechanism of D-TENG. **c** Open-circuit voltage for different droplet release heights. **d** Comparison of the voltage generated from moisture or fog in this work compared to refs. [32–39]. **e** Number of droplets generated in 1 h (combination of four parallel arrays) and the corresponding open-circuit voltage under different fog flows. **f** Working mechanism of the single charge generator and its transfer charge. **g, h** Open-circuit voltage and transfer charge of the negative charge generator.

electrode on the FEP, the accumulated charge is transferred from the lower electrode to the interface between the droplet and polymer. Subsequently, the triboelectric charge returns to the lower electrode as the droplet leaves[30,31]. The as-fabricated devices presented a stable surface charge of 42.6 nC after approximately 770 droplets (Supplementary Figs. 17 and 18), showing a fast precharge capability. Subsequently, parameters such as the impact angle, droplet volume, and droplet release height were optimized (Fig. 5c, Supplementary Notes 4, 5, and Supplementary Figs. 19, 20). Interestingly, an open-circuit voltage can reach 103.2 V, which could light 400 commercial LED bulbs (Fig. 5d and Supplementary Movie 1) and). Its output performance was much higher than that of related research work that generated electricity from moisture or fog (Fig. 5d)[32–39].

To verify the power generation from fog, the water harvesting and power generation units were assembled on a self-made platform, and different fog flows were simulated using a humidifier (Fig. 5a). At different fog flows, although the number of droplets produced in 1 h increased from 94 to 179, there was little change in the voltage because of the similar volume of droplets passing through the droplet controller (Fig. 5e).

To develop a negative charge generator, a diode was connected in parallel between the upper and lower electrodes, the lower electrode was grounded, and the upper electrode was connected to a conductive cloth tape. Because the droplet contacts the upper electrode, the negative charge is transferred to the conductive cloth tape. Subsequently, the positive charge returned to the lower electrode through the diode as the droplet left (Fig. 5f and Supplementary Fig. 21). As expected, the average open-circuit voltage and transfer charge in a single cycle was 90.6 V and 43.6 nC, respectively, demonstrating the high efficiency of the negative charge generator (Fig. 5g, h).

**Electrostatically assisted fog collection.** Based on the principle of triboelectrification, FEP accumulates and stores electrons as the water droplets slip, which causes electrostatic adsorption to fog (Fig. 6ai)[40,41]. Therefore, the negative charge generator was externally attached to the back of the hydrophobic channel for constant charge replenishment (Fig. 6a). It is hypothesized that the droplet would be preferentially condensed and enlarged on charged regions and merge with droplets on the spine, which

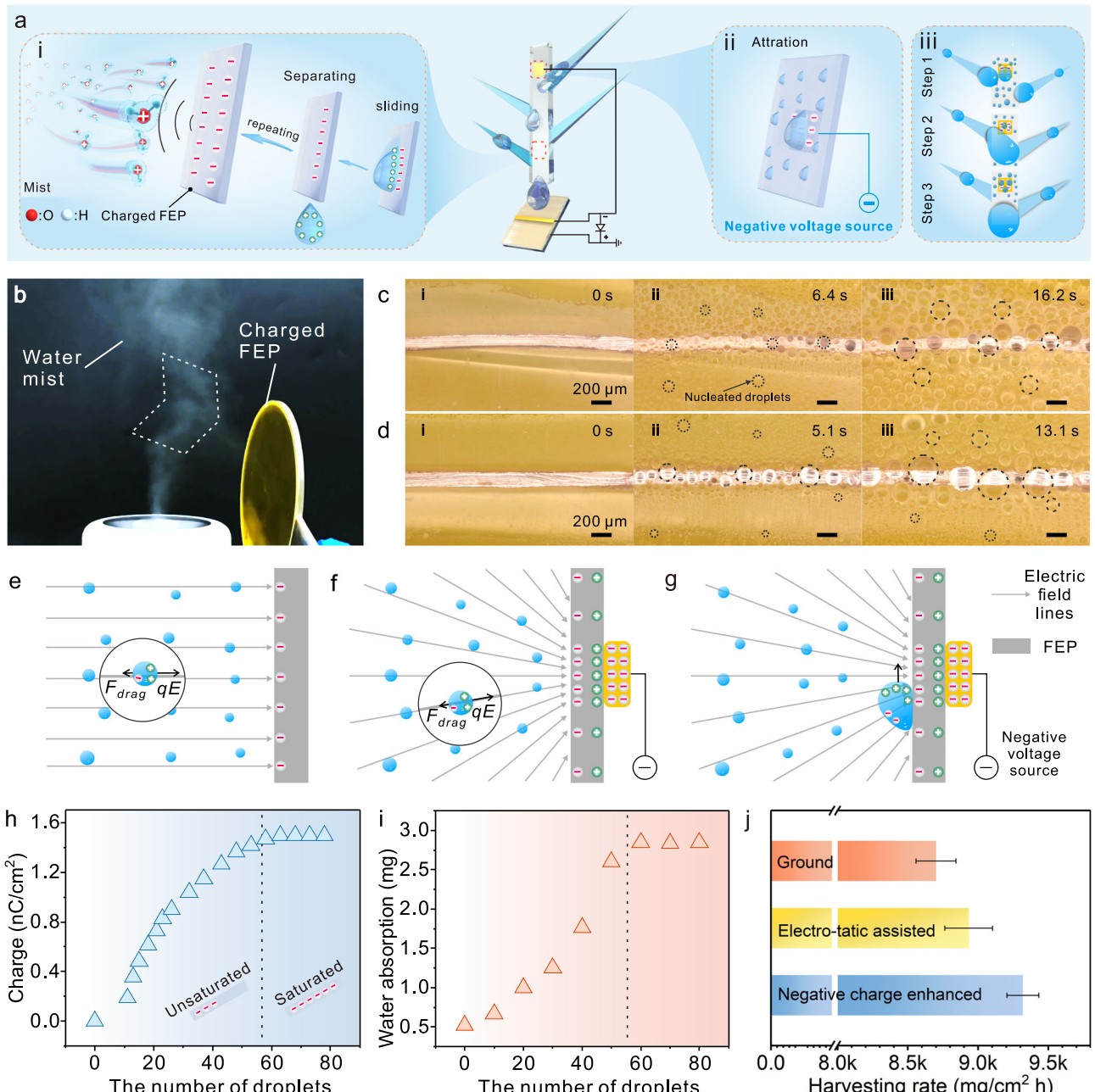

**Fig. 6 Electrostatically assisted water collection. a** Schematic diagram of electrostatically assisted water harvesting. Static electricity was from the triboelectrification between droplets and FEP as well as external charges. **b** Upward moving water mist was attracted to the charged FEP wafer. A copper wire was attached to the back of the FEP, which was subsequently exposed to fog remaining uncharged **c** and charged **d**. **e** Schematic of droplets in an electrostatic field. **f, g** Schematic of the droplets in the electric field applied charge. **h, i** Charge transfer and water absorption of a thin FEP strip (5 mm × 50 mm) as water droplets slide-off. **j** Water-harvesting rate undergrounding, self-triboelectrification, and external charge.

would facilitate the transfer and shedding of pinned droplets (Fig. 6aii–iii). Figure 6b shows the attractive effect of charged FEP on the water mist. After the FEP wafer was charged by the droplets or filter paper, the water mist drifted to the FEP wafer, followed by adhesion due to electrostatic adsorption (Supplementary Movie 2). Moreover, a piece of FEP film with electrodes attached to the back was exposed to water mist to investigate the effect of external charging on mist adsorption. As can be observed in Fig. 6c, with the increase in exposure time, tiny water droplets appeared on the EFP, followed by growth. The water droplets were uniformly distributed and were similar in size. After a negative charge injection, larger droplets appeared in the region where the electrodes were attached (Fig. 6d). The above

phenomenon may be explained by the strong Coulomb force. It is assumed that the droplets do not collide with each other and that the electric field does not change as the droplets attach. As an electric field (**E**) generated after the FEP was charged, the motion state of the droplet in the electric field could be calculated by the formula 3[8],

$$m\frac{d\mathbf{u}}{dt} = 6\pi\eta_g R_d(\mathbf{w} - \mathbf{u}) + q\mathbf{E} \tag{3}$$

where $m$ is the mass of the droplet, $\mathbf{u}$ is its velocity, $R_d$ is its radius, $q$ is its charge, $t$ is the time, $\eta_g$ is the air viscosity, and $\mathbf{w}$ is air velocity. The droplets were triboelectrically positive due to friction with the air, and the positive charges would be rearranged

by polarization effects of the electrostatic field[8,18]. Therefore, the droplet was mainly driven by the Coulomb force (to overcome the drag resistance $F_{drag}$), moved along the electric field lines towards the FEP, and finally attached to the FEP surface (Fig. 6e). After the negative charge injection, the locally high charge density would lead to an inhomogeneous electric field. The droplets would move towards the FEP, and enrich in regions with high charge density (Fig. 6f). In addition, agglomerated droplets on the FEP surface would also be driven toward this region[42] (Fig. 6g).

Subsequently, the transfer charge on an FEP strip (simulating a hydrophobic channel) as the droplet slipped was recorded. The transfer charge increased from 0.18 to 1.50 nC/cm$^2$ and remained constant with increasing time (Fig. 6h). Simultaneously, the fog adsorption capacity of the FEP strips was investigated. It can be seen from Fig. 6i, with the increase in the droplet numbers, the fog adsorption increased from 0.52 to 2.85 mg and then remained stable. This trend is consistent with the results of the surface charge transfer. Furthermore, the water-harvesting rate under-grounding, self-triboelectrification, and the external charge was tested (Fig. 6j). Interestingly, a high water-harvesting rate of 93.18 kg/m$^2$ h was obtained from external charging using a D-TENG, which is higher than that of current fog collectors mimic beetles and cactus (Supplementary Table 1)[14,43–50].

## Discussion

In this study, an asymmetric amphiphilic surface inspired by cactus spines and beetle elytra was designed to achieve high water collection efficiency (93.18 kg/m$^2$ h) by combining spontaneous interfacial triboelectric adsorption. The efficient water-harvesting rate was mainly owing to the synergistic effect of the hydrophilic components (droplet nucleation sites) and hydrophobic components of the amphiphilic cellulose esters, as well as the Laplace pressure from asymmetry. Furthermore, the spontaneous electrostatic adsorption generated between the droplet and water collector as well as the electrostatic adsorption enhanced by the applied charge was demonstrated. Ultimately, such a system is not only suitable for foggy areas but is also expected to be applied to the steam recovery and energy generation of cooling towers in thermal power plants and paper mills, which furnishes a universal solution for mitigating the water-energy nexus.

In the development of the water-harvesting system, only a simple compact model was fabricated to demonstrate the coupling of the water-harvesting unit and the power generation unit. Since the skeleton material of the homemade platform could be replaced and upgraded, the key to the robustness of the system lies in the amphiphilic asymmetric surface. Surprisingly, even exposed to a wind of 16.2 m/s (equivalent to a moderate gale) for 5 min, the amphiphilic asymmetric surface was slightly deformed, but not cracked (Supplementary Table 2), demonstrating good wind resistance of the water-harvesting device. For practical applications, it could be improved from the aspects of dynamics, mechanics, electricity, and so on. For example, an all-steel skeleton structure could be upgraded to enhance the robustness under extreme conditions. Energy storage devices and circuit management units could be added to improve the utilization rate of electricity. After the electrical properties of the D-TENG are calibrated, wireless sensing units could be added to detect components of the collected water in real-time.

## Methods

**Preparation and characterization of ACEC**. Dried MCC (1 g) was suspended in anhydrous pyridine (40 mL). After heating this suspension to 70 °C, 5.33 mL of 10-undecenoyl chloride was added under magnetic stirring. The stirring process was continued for 3 h at 70 °C. Subsequently, the suspension was poured into 300 mL of ethanol for precipitation, followed by high-speed centrifugation (8000 rpm). Pure ACEC was obtained after multiple purifications.

To characterize the chemical structure of ACEC, an elemental analyzer (Vario EL cube, Elementar, Germany), a solid-state magnetic spectrometer (Avance III 400 MHz, Bruker, Switzerland), Fourier transform infrared (FTIR) spectrometer (TENSOR II, Bruker, Germany), and X-ray photoelectron spectroscopy (XPS, ESCA Lab 250Xi, Thermo Fisher Scientific) were used.

**Preparation of asymmetric amphiphilic surface**. The FEP film (150 μm) was cut according to the sketch using a laser engraver (VLS3.50-SYS, Universal, USA), followed by a single-sided coating of ACEC in the solution (25 mg/mL) using a pipette (10 μL). After drying at room temperature, a double-biomimetic water-harvesting structure was obtained. The morphology of the FEP and ACEC@FEP spines was observed using a scanning electron microscope (Sigma 300, ZEISS, Germany) and an industrial microscope (VHX-6000, Keyence, Japan). The wettability and surface free energy (OWRK method) were verified using a contact angle meter (DSA35, Kruss, Germany). The adhesion strength of the ACEC was measured by using a micron scratch instrument (CSM-MCT, Switzerland) six times.

**Fabrication of D-TENG and self-made platform**. The commercial double-sided conductive cloth tape (as the lower electrode) was bonded to acrylic plates (60 mm × 70 mm), and then the FEP film was attached (25 μm) to the conductive cloth tape. After arranging the copper foil (1 mm × 6 mm × 0.1 mm) as the upper electrode, a droplet generator module was obtained. The negative charge generator was obtained by connecting the diodes in parallel and then grounding the anode. To enhance the fog collection, a copper wire (0.3 mm × 10 mm) was bonded to the back of the FEP hydrophobic channel (behind the top set of spines) and then connected to the negative terminal of the negative charge generator. All of the above parts were assembled on a homemade acrylic skeleton.

**Electric measurement**. A commercial drip device was employed to simulate the dripping of the collected liquid (the volume of the droplet was approximately 96 μL by adjusting the diameter of the plastic nozzle. If not specified, all the water was deionized water). When the droplets impinged on the D-TENG, the open-circuit voltage and transfer charge were recorded using a programmable electrometer (Keithley 6514) at 19 °C, humidity of 60%. To optimize the parameters of D-TENG, the release height of the droplet was adjusted to increase from 5 to 20 cm, the impacting angle was adjusted to increase from 15 to 75° impact angle, and the droplet volume was adjusted to increase from 54 to 96 mL. The fog flow was adjusted to increase from 100 to 300 mL/h, and the open-circuit voltage of the droplets generated by the combination of the four basic units was recorded (release height: 20 cm). During the optimization of experimental parameters, the D-TENG was pre-charged using ion injection (Zerostat III, Milty) to reduce the time to reach stability.

**Fog harvesting**. A fog generator (YC-D205, Yadu) was used to simulate the fog environment with a fog flow of 300 mL/h and a constant speed of 0.3 m/s. The amphiphilic asymmetric surface was placed in parallel at a distance of 5 cm from the fog outlet, and a weighing bottle was placed directly below it to collect the resulting droplets. The quality of the collected liquid was recorded every 10 min. By adjusting the fog flow from 100 to 300 mL/h, the volume and number of droplets generated by the combination of the four basic units in 1 h were recorded. To investigate the fog adsorption from triboelectrification, an FEP wafer (diameter:13 cm) was charged by the droplets impinging or the friction of the filter paper and then placed close to the water mist. To observe fog adsorption when an external charge was applied, the negative charge generator was externally attached to the back of the hydrophobic channel for constant charge replenishment (the frequency of droplets impinging: 1 Hz). The dynamic process of fog collection was observed using a high-speed camera (AX200, Photron, Japan) and an industrial microscope. In all of the above operations, the relative humidity around the amphiphilic asymmetric surface was maintained at 99.9% (Supplementary Fig. 22).

### Data availability

All data generated in this study are provided in the Supplementary Information/Source Data file. Source data are provided with this paper.

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

## Acknowledgements

This research was supported by the National Natural Science Foundation of China (31971604, S.N.) and the Natural Science Foundation of Guangxi Province (2018GXNSFDA281050, S.N.).

## Author contributions

S.Z. and S.N. conceived the idea and designed the study. S.W. provided scientific guidance throughout. S.Z. and M.C. designed ACEC and biomimetic structures. Y.L., T.L., and Q. F. characterized the materials. J.M. fabricated D-TENG. J.W., Y.Q., and B.L. performed the electrical measurement and analyzed the data. S.Z. and S.N. wrote the paper. All authors discussed the results and commented on the manuscript.

## Competing interests

The authors declare no competing interests.
