## [Peer Review File · Nature Communications]

Bioinspired asymmetric amphiphilic surface for triboelectric enhanced efficient water harvestingREVIEWER COMMENTS

Reviewer #1 (Remarks to the Author):

The manuscript entitled "Bioinspired asymmetric amphiphilic surface for triboelectric enhanced efficient water harvesting" presents a novel combination of bioinspired self-driven water harvesting and interfacial triboelectric charge assisted electrostatic fog adsorption, showing a markable performance as a water harvesting system. The manuscript can be considered for publication after addressing the following issues,

1. The humidity data for all the fog collection experiments should be provided. Otherwise, the claim "unprecedented water harvesting efficiency (9318 mg/cm² h)" is not convincing.
2. Page5 Line102-103. Why the trend of data becomes sparse in Fig.2j? What is the effect of this change on the network structure?
3. There are some subfigures (such as Fig 2f, 5d, and 5e) not being cited in the article. Are these data not necessary? If they make sense, please clarify and cross-reference them in the article.
4. P9-P10. It is necessary for readers to know how the transporting time of droplets is calculated. What are the specific criteria for evaluating the transport rate of droplets?
5. P9-P10. The explanation for the higher water harvesting efficiency at 60 ° inclined angle is not convincing enough. Under the premise of the longer transporting time, why can a high proportion of spine area within the same height range lead to higher water harvesting efficiency? This needs to be explained in further detail.
6. P9-P10. For the exploration of angle change, it is necessary to further explain the water harvesting efficiency for the 90 ° case, especially whether the efficiency after 60 ° behaves a downward trend.
7. P9. For the experiment of the fit width of the transport channel, more width experimental groups between 1-3mm are needed to prove that the optimal harvesting efficiency can be obtained at 2mm. And for Fig. S6, for the text "a decrease or increase in width resulted in a delay in the droplet movement", a more detailed explanation in mechanism is suggested.
8. I recommend that the parameter symbols in Fig. 4f be defined to indicate "The ipsilateral and contralateral vertical distances of the spine". Otherwise, the instructions in this article are difficult to be understood.
9. P11 Line 225. What is the purpose of ion injection? The relationship between ion injection and the research about D-TENG should be clarified. Moreover, how many LEDs can D-TENG light without ion injection?

Reviewer #2 (Remarks to the Author):

In the manuscript of "Bioinspired asymmetric amphiphilic surface for triboelectric enhanced efficient water harvesting", the authors proposed an asymmetric amphiphilic surface combining notable features of cactus spines and elytra of Namib Desert beetles for triboelectric-enhanced water harvesting. In general, the story is interesting. However, more information or experiments are required to be carried out in order to improve the quality of this manuscript..

1. In the part of design of system design, the authors fabricated the biomimetic cactus arrays by using laser engraved fluorinated ethylene propylene (FEP) membranes for Laplace driving forces. However, there is no description about how Laplace driving forces influence the efficiency of harvesting water or comparison with symmetric spine. The authors may need to provide more details for this experiment.
2. Some minor revisions need to be checked throughout the manuscript. For example, in the Fig. 3a, it's "Contact angle" instead of "Contanct angle".
3. To evaluate the robustness of the coatings, droplet sliding as well as immersion were performed to observe the morphology and contact angle changes. However, in the figure of 3d and 3e, it's hard to tell whether the coating of ACEC is peel off or not. Optical microscope or advanced images are recommended to add in the figures to convince the readers.
4. For fog harvesting, a fog generator was used to simulate the fog environment. It's worth to noted that in real case, the fog in the air may be affected by the environment condition, especially humidity. The authors should better provide some data to verify the relationship of fog flows and humidity.

5. In the output performance of D-TENG, ion injection technique was applied to enhance the output voltage. Would it influence the surface wettability of FEP and how long it could be sustained? The authors may need to provide some data to prove the stability of triboelectric effect.

6. For electrical measurement, approximately constant 96 μL droplet impinged on the D-TENG and investigating the output performance by adjust different droplet release heights. Also, vary volume of droplet may and the impact angle may change the output. I would like to suggest the authors can add some statement about this.

Point-by-point Response to Reviewers' Comments

(Manuscript ID: nn-2021-01490a)

Reviewer: #1

The manuscript entitled “Bioinspired asymmetric amphiphilic surface for triboelectric enhanced efficient water harvesting” presents a novel combination of bioinspired self-driven water harvesting and interfacial triboelectric charge assisted electrostatic fog adsorption, showing a markable performance as a water harvesting system. The manuscript can be considered for publication after addressing the following issues.

Response: We are truly gratefully for the review’s generous comments on our work as “*a markable performance as a water harvesting system.*” and the valuable feedback that we have used to improve the quality of the manuscript. We strongly agree with the reviewer’s suggestions and thank the reviewer again.

1. The humidity data for all the fog collection experiments should be provided. Otherwise, the claim “unprecedented water harvesting efficiency (9318 mg/cm² h)” is not convincing.

Response: (1) We greatly appreciate the reviewer’s professional comment and valuable question. We strongly agree with the reviewer that the humidity data are very important for the fog harvesting. Related humidity data are as follows.

① A test chamber was designed to minimize the impact of the environment (such as wind). Additionally, a high-precision hygrometer was employed to record the relative humidity around the amphiphilic asymmetric surface (Figure R1).

Figure R1. Optical photograph of the humidity test setup.

② The relative humidity at different fog flow rates was recorded and has been added in the revised manuscript (Fig. S22). As shown in Fig. S22, the relative humidity around the amphiphilic asymmetric surface could reach 99.9% successively as the fog flow rate increased from 100 mL/h to 300 mL/h. The manuscript has been revised according to the reviewer’s suggestions and the detailed description is as follows.

“In all of the above operations, the relative humidity around the amphiphilic asymmetric surface was maintained at 99.9% (Fig. S22).”

Supplementary Figure 22. As the fog flow rate increased from 100 mL/h to 100 mL/h, the relative humidity around the amphiphilic asymmetric surface could reach 99.9% successively (ambient temperature $\sim 25^{\circ}\text{C}$, ambient humidity $58 \pm 1\%$).

(2) We apologize for the inaccurate description. As the reviewer stated that *Nature Communications* seeks for a sober writing style, avoiding “phrases of primacy” such as “unprecedented”. Thus, we checked the manuscript carefully and removed the claims of unprecedented throughout the paper, and we hope to receive your approval. Details are as follows.

“and further enhancing the electrostatic adsorption utilizing external charges, which achieved a water harvesting efficiency of $93.18 \text{ kg/m}^2 \text{ h}$.”

2. Page5 Line102-103. Why the trend of data becomes sparse in Fig.2j? What is the effect of this change on the network structure? Page5 line102 – 103.

Response: We apologize for the lack of detailed description, which makes it difficult for readers to obtain useful information. In fact, this phenomenon can be attributed to the coating process of ACEC. There is an energy barrier at the junction between the spine and the hydrophobic channel. To reduce the energy barrier for droplet crossing, besides grafting non-polar groups to reduce the surface energy of ACEC, the coating amount of ACEC at the spine root was reduced by progressive multiple coating. The details are shown in Fig. R2. Firstly, a layer of ACEC was coated on the entire surface of the spine, and after it was dry, the second layer was applied at about two-thirds of the distance from the spine tip, and then the third layer was applied. Through this progressive method, the content of ACEC on the FEP was gradually reduced, showing relative sparse phenomenon at the root of the spine. We are grateful to the reviewers for pointing out inaccurate descriptions in our manuscript. We have carefully revised the manuscript and hope to receive your approval. Details are as follows.

“With the gradual increase of the spine width, the spine presented as a prismatic table with the upper and sides coated with ACEC (Fig. 2j-k).”

Figure R 2: Schematic diagram for the preparation method of ACEC@FEP.

3. There are some subfigures (such as Fig 2f, 5d, and 5e) not being cited in the article. Are these data not necessary? If they make sense, please clarify and cross-reference them in the article.

Response: We apologize for the confusion to the reviewer due to the inconsistency of the order of the discussion and the pictures. Accordingly, the order of pictures has been adjusted and the related descriptions have been revised. The details are as follows.

“In addition, compared with the smooth surface of MCC, ACEC shows a porous structure, which provides more gas-solid interface (Fig. 2f).”

“The effect of the droplet release height on the output performance was investigated. As the release height increased, the open-circuit voltage increased from 59.4 V to 103.1 V (Fig. 5c).”

“there was little change in the voltage because of the similar volume of droplets passing through the droplet controller (Fig. 5e).”

Fig. 5. Output performance of the D-TENG. **a** Self-made platform for fog collection and power generation. **b** Schematic diagram of the working mechanism of D-TENG. **c** Open-circuit voltage for different droplet release height. **d** Comparison of the voltage generated from moisture or fog in this work with previous reports. **e** Number of droplets generated in 1 h (combination of 4 parallel arrays) and the corresponding open-circuit voltage under different fog flow rates. **f** Working mechanism of the single charge generator and its transfer charge. **g, h** Open-circuit voltage and transfer charge of the negative charge generator.

4. P9-P10. It is necessary for readers to know how the transporting time of droplets is calculated. What are the specific criteria for evaluating the transport rate of droplets?

Response: We highly appreciate the reviewer’s professional comments, which kindly help us to further improve our work.

(1) To further investigate the mechanism of the fog collection, taking a single spine as an example, the droplet transmission during the harvesting is divided into three stages in detail:

- t₁: Time for water droplets creep from the spine tip to the root;
- t₂: Time for droplets grow and then cross into hydrophobic channels;
- t₃: Time for the droplet hangs at the junction until falls off.

“Compared with FEP, ACEC@FEP exhibited faster fog collection and transport efficiency ($t_1=13.3$ s) (Fig. S7-8, Fig. 4a).”

Supplementary Figure 7. Taking a single spine as an example, the droplet transmission during the harvesting is divided into three stages in detail: t₁: Time for water droplets creep from the spine tip to the root; t₂: Time for droplets grow and then cross into hydrophobic channels; t₃: Time for the droplet hangs at the junction until falls off.

(2) So far, the research on fog and water harvesting has made great progress, such as, Shi et al. prepared a hydrogel film with a three-dimensional tree-like microstructure for fog harvesting. They described a water transport cycle including fog droplets nucleation followed by their transport, growth, and eventual drainage. Li et al. tailoring the aerodynamics of fog-laden wind using kirigami structures for water harvesting. They decoupled the fog harvesting process into two separate stages: an array of curvature-induced vortices to promote the interception of incoming droplets, and the asymmetric and connected folds to directionally gather the deposited droplets. For various fog collectors, the definition of droplet transport time is different due to different structural designs and droplet transport modes. The indicator is usually used to guide the optimization of a certain experimental parameter. For water harvesting, the water harvesting efficiency can be used for comparison, that is, the mass of water collected by the area/mass of the fog collector per unit time.

5. P9-P10. The explanation for the higher water harvesting efficiency at 60 °inclined angle is not convincing enough. Under the premise of the longer transporting time, why can a high proportion of spine area within the same height range lead to higher water harvesting efficiency? This needs to be explained in further detail.

Response: We sincerely thank the reviewer’s professional question and valuable suggestion. According to the reviewer's comments, we have realized that the explanation for the higher water harvesting efficiency at the angle of 60° is indeed insufficient. To explain this phenomenon, a series of experiments were supplemented, the results show that the higher water harvesting efficiency at the angle of 60° may be due to the combined effect of the inclination angle, the area ratio of the spine and the length of the borderline between the spine and the hydrophobic channel (Fig. S7, S11-S13).

(1) Fig. 4b illustrates that droplets could creep into the hydrophobic channel from the spine at different angles. And the corresponding time from the start of harvesting until the droplet crosses into the hydrophobic channel (t_1+t_2). Such a time mainly depends on the tilt angle. According to the force analysis, the additional driving force provided by the gravitational component decreased as the inclination angle increases, which will lead to a gradual increase of t_1+t_2 (Fig. S11)

Supplementary Figure 7. Taking a single spine as an example, the droplet transmission during the harvesting is divided into three stages in detail: t_1 : Time for water droplets creep from the spine tip to the root; t_2 : Time for droplets grow and then cross into hydrophobic channels; t_3 : Time for the droplet hangs at the junction until falls off.

Supplementary Figure 11. During the water harvesting, t_1+t_2 increased as the angle increased, and t_3 decreased gradually as the angle increased.

(2) When the droplet crept into the hydrophobic channel, it was inevitably dragged due to the hydrophilicity of ACEC, resulting in prolonged shedding time (t_3). It can be

seen from Fig. S12 that the length of the borderline decreased from 2.02 mm to 1 mm with increasing angle. Notably, t_3 decreased with decreasing contact line (increasing angle) (Fig. S12), indicating that a shorter borderline (larger angle) facilitates rapid detachment of the droplet.

Supplementary Figure 12. The length of the borderline decreased from 2.02 mm to 1 mm with increasing angle (The distance between the midpoint of the borderline and the tip of the spine is 10 mm).

(3) The spine primarily capture fog and dominate the entire water harvesting process. In general, the more spines at the same height (the larger the corresponding area ratio), the higher efficiency of water harvesting. The length of the boundary line decreases with the increase of the angle, thereby under the same number of spines and vertical spacing, the area ratio of the spine and corresponding height are positively and negatively correlated with the size of the inclination angle, respectively. (Fig. S13, Fig. R3).

Supplementary Figure 13. The ratio of spine area corresponding to different inclination angles (The distance between the midpoint of the junction line and the tip of the spine is 10 mm).

Figure R3. The hydrophobic channel height decreased with increasing inclination angle (vertical distance between spines is 2 mm).

To sum up, the larger additional driving force and shorter borderline as well as larger area ratio of the spine together result in the best water harvesting efficiency at 60°. The details are as follows.

“As expected, even angle between the spine and the hydrophobic channel increased from 30° to 90°, the droplet could cross into the hydrophobic channel (Fig. 4b). The force analysis showed that when the spine is horizontal, the droplet is subjected to the Laplace pressure (F_L) toward the root, downward gravity (G), and opposite adhesion forces (F_{V1} , F_{V2}). When the spine is inclined, G can be decomposed into F_y and F_x , and F_x is coaxial with F_L (providing an additional driving force for the droplet) and decreased with an increase in the vertical angle (Fig. 4c), which leads to a decrease of the droplet transport speed. Nevertheless, there is a borderline between the spine and the hydrophobic channel. When the droplet crept into the hydrophobic channel, it would inevitably be dragged due to the hydrophilicity of ACEC, resulting in prolonged shedding time (t_3). Additionally, since the spines dominate the water harvesting, more spines within the same height (i.e., higher ratio of spine area) are beneficial for water harvesting (Fig. S13). Therefore, a larger additional driving force and shorter borderline as well as higher spine area ratio together result in the highest water harvesting efficiency at 60°.”

6. P9-P10. For the exploration of angle change, it is necessary to further explain the water harvesting efficiency for the 90 ° case, especially whether the efficiency after 60 ° behaves a downward trend.

Response: (1) We are grateful for the reviewer’s professional suggestions and strongly agreed with the reviewer’s suggestions. Accordingly, we observed the process of fog collection on an amphiphilic asymmetric surface with the angle of 90°. As shown in Fig. 4bvii–viii, the condensed droplet could cross the borderline into the hydrophobic channel. This phenomenon indicates that droplet could creep into the hydrophobic channel at different angles, which further verifies the feasibility of the design idea of amphiphilic asymmetric surfaces.

Fig. 4bvii-viii. Fog harvesting in an array with the angle of 90°.

(2) To further explore the effect of the angle on water harvesting, the water harvesting efficiency of the amphiphilic asymmetric surface with different angles was recorded. As shown in Fig. 4d, the water harvesting efficiency gradually decreased after 60°. The manuscript has been revised and we hope to receive your approval. The details are as follows.

Fig. 4d. Water harvesting rate of the array with different angles.

7. P9. For the experiment of the fit width of the transport channel, more width experimental groups between 1-3mm are needed to prove that the optimal harvesting efficiency can be obtained at 2mm. And for Fig. S6, for the text “a decrease or increase in width resulted in a delay in the droplet movement”, a more detailed explanation in mechanism is suggested.

Response: We greatly appreciate for reviewer’s professional comments. The reviewer’s comments are of great help to our work.

(1) We investigated the water harvesting efficiency of the amphiphilic asymmetric surface with hydrophobic channel width between 1-3 mm. As can be seen from Fig. 4e, with the increase of the hydrophobic channel width, the water harvesting efficiency gradually increased and then decreased, reaching a peak when the width is 2 mm. The relevant content has been revised in the manuscript (Fig. 4e).

Fig. 4e. Water harvesting rate of the array with different hydrophobic channel widths.

(2) To prove the effect of the hydrophobic channel width on the droplet transport, the dynamic process of droplets at different width was investigated (Fig. S14). As can be observed, the merged droplets hung at the junction of the narrower hydrophobic channel due to excessive solid-liquid contact lines. As the width increased, the hanging time of the droplet decreased. Nevertheless, wider hydrophobic channels presented larger droplet coalescence distances, which would delay or hinder droplet coalescence. Therefore, a decrease or increase in width resulted in a delay in the droplet movement. The manuscript has been revised and we hope to receive your approval. The details are as follows.

“The optimal water harvesting rate was obtained with a width of 2 mm, and a decrease or increase in width resulted in a delay in the droplet movement due to excessive solid-liquid contact lines or inefficiency/ineffective droplet coalescence (Fig. 4e, Fig. S14, and Fig. S15).”

Supplementary Figure 14. The dynamic process of the droplet at different hydrophobic channel width (1-3 mm, tilt angle=60°). As can be observed, the merged droplets hung at the junction of the narrower hydrophobic channel due to excessive solid-liquid contact lines. As the width increased, the hanging time of the droplet

decreased. Nevertheless, wider hydrophobic channels presented larger droplet coalescence distances, which would delay or hinder droplet coalescence. Therefore, a decrease or increase in width resulted in a delay in the droplet movement.

Supplementary Figure 15. Schematic diagram of fog harvesting with smaller or larger width of the transport channel.

8. I recommend that the parameter symbols in Fig. 4f be defined to indicate “The ipsilateral and contralateral vertical distances of the spine”. Otherwise, the instructions in this article are difficult to be understood.

Response: We highly appreciate the reviewer’s professional comments, which kindly help us to further improve our work. According to the reviewer’s suggestion, we have revised in the manuscript. The details are as follows.

Fig. 4f Water harvesting rate of an array with different hydrophobic channel widths and different vertical distances of the spines; the root width of the spine is 1.2 mm.

9. P11 Line 225. What is the purpose of ion injection? The relationship between ion injection and the research about D-TENG should be clarified. Moreover, how many LEDs can D-TENG light without ion injection?

Response: We highly appreciate the reviewer’s professional comments and we are pleased to have the opportunity to answer the questions.

(1) Notably, it takes too much time to accumulate surface charge to saturation

through continuous droplet impacting. Therefore, we injected negative ions (CO_3^- , NO_3^- , NO_2^- , O_3^- and O_2^-) generated by air ionization onto FEP, the surface charge rapidly increased to a higher level, even beyond saturation. Accordingly, the start-up time of the D-TENG could be effectively shortened, and the initial output performance could also be improved to some extent. In our manuscript, the ion injection is mainly used to shorten the time for D-TENG to reach stability.

(2) To clarify the relationship between ion injection and D-TENG in detail, we recorded the transferred charge of D-TENG undergoing ion injection and self-charging. As shown in Fig. S17, to achieve a stable electrical output of D-TENG, at least 770 consecutive droplet impacting were required. After ion injection (a high electrical output could be obtained in a short period, 500 LED lamps could be instantly turned on), the transferred charge fell back to the stable state after nearly 300 droplets impacting (Fig. S18). That is, the method of ion injection enables D-TENG to reach stability quickly.

Supplementary Figure 17. As the droplets continue to impinge on the D-TENG, the amount of charge on the FEP surface increased gradually followed by remaining stable.

Supplementary Figure 18. After ion injection on the D-TENG surface, the transfer charge decreased rapidly from 371 nC followed by remaining stable.

(3) According to the reviewer's suggestion, we explored the number of lit bulbs without ion injection. As shown in Fig. R4, the stable D-TENG could light up 400 LED lamps continuously. It is worth noting that the brightness of the bulb looks different at different angles due to the directional emitting of LEDs.

Figure R4. Photo of 400 LED bulbs lit by D-TENG after droplet impacting.

The details are as follows.

“During the optimization of experimental parameters, the D-TENG was pre-charged using ion injection (Zerostat III, Milty) to reduce the time to reach stability.”

Reviewer: #2

In the manuscript of “Bioinspired asymmetric amphiphilic surface for triboelectric enhanced efficient water harvesting”, the authors proposed an asymmetric amphiphilic surface combining notable features of cactus spines and elytra of Namib Desert beetles for triboelectric-enhanced water harvesting. In general, the story is interesting. However, more information or experiments are required to be carried out in order to improve the quality of this manuscript.

Response: We are deeply grateful for the review’s positive comments on our work as *“the story is interesting.”*. We also highly appreciate the valuable feedback, which has provided a great help to our work. To improve the quality of our manuscript, the effect of the spine with different aspect ratios and Laplace pressure on water harvesting was further investigated. The water harvesting efficiency of symmetric spines and asymmetric spines was compared. Scanning electron microscopy was added to observe the surface topography of the ACEC@FEP after the droplet sliding down and immersion in water. Relative humidity data were recorded under different fog flow rates. The wettability (static contact angle and dynamic contact angle) of FEP after ion injection was investigated. The effects of the impact angle and droplet volume on the output performance of the D-TENG were carried out. We hope to receive the reviewer’s approval and the details are as follows.

1. In the part of design of system design, the authors fabricated the biomimetic cactus arrays by using laser engraved fluorinated ethylene propylene (FEP) membranes for Laplace driving forces. However, there is no description about how Laplace driving forces influence the efficiency of harvesting water or comparison with symmetric spine. The authors may need to provide more details for this experiment.

Response: We greatly appreciate the reviewer’s professional comments, which are of

great help to our work. Accordingly, the water harvesting efficiency and $\sin\alpha$ of asymmetric/symmetric structures (different heights and widths) was added in the revised manuscript, and we hope to receive your approval.

(1) The Laplace pressure (ΔP) of the asymmetric surface can be calculated according to formula S3

$$\Delta P = \frac{dP}{dz} \Big|_{\Omega} = - \frac{2\gamma}{(r + R_0)^2} \sin\alpha \quad (\text{S3})$$

where Ω is the droplet volume, γ is the surface tension of the droplet, r is the surface local radius, R_0 is the droplet radius, and α is the half vertex angle. It can be seen from the formula that ΔP increases as $\sin\alpha$ decreased. Therefore, a small $\sin\alpha$ contributes to a large ΔP . It can be seen from Fig. S9 that the water harvesting efficiency of asymmetric surfaces with different widths and heights increased with the decrease of $\sin\alpha$. It is worth noting that, due to limitations in instrument accuracy and material properties, the smallest $\sin\alpha$ was obtained at a height of 10 mm and a width of 1 mm.

Supplementary Figure 9. The water harvesting efficiency and $\sin\alpha$ of asymmetric/symmetric structures. The Laplace pressure (ΔP) of the asymmetric surface can be calculated according to formula S3.

(2) To compare the water harvesting efficiency of asymmetric and symmetric structures, more details for this experiment were added (spines with the height of 10 mm and width of 1 mm). As shown in Fig. S10, the water harvesting efficiency of the symmetric structure was lower than that of the asymmetric. Therefore, an asymmetric structure with a height of 10 mm and a width of 1 mm was chosen.

Supplementary Figure 10. The water harvesting efficiency of symmetric and asymmetric structure. Notably, the asymmetric was greater than that of the symmetric.

Therefore, an asymmetric structure with a height of 10 mm and a width of 1 mm was chosen.

Details are as follows.

“According to Laplace formula (S3), spines with different *sina* were prepared by adjusting the height and width, and their water harvesting efficiency was investigated (Fig. S9 and S10). Compared with the symmetric structure, the asymmetric presented a higher water harvesting efficiency. Especially at a height of 10 mm and a width of 1 mm resulted in the smallest *sina* value and the best harvesting efficiency”

2. Some minor revisions need to be checked throughout the manuscript. For example, in the Fig. 3a, it’s “Contact angle” instead of “Contanct angle”.

Response: We apologize for the writing mistake and highly appreciate the reviewers for their careful review. We have carefully checked the full manuscript and found an erroneous legend in Fig. 3c. Afterwards, the Fig. 3a and Fig. 3c have been revised and we hope to receive your approval. Details are as follows.

Fig. 3. Wettability and robustness. a Photographs of contact angle of MCC, ACEC and FEP. b Surface free energy of FEP and ACEC. c Advancing and receding contact angles of FEP and ACEC@FEP. d Photographs of ACEC@FEP surface after the water droplets sliding down. e Contact angle and photographs after immersion in water for different time. f Adhesion strength between ACEC and FEP.

3. To evaluate the robustness of the coatings, droplet sliding as well as immersion were performed to observe the morphology and contact angle changes. However, in the figure of 3d and 3e, it’s hard to tell whether the coating of ACEC is peel off or not. Optical microscope or advanced images are recommended to add in the figures to convince the readers.

Response: We greatly appreciate the reviewer’s valuable question, which really helps us to improve our work. We realized that it is difficult to observe the microscopic topography of the ACEC@FEP using optical photographs. According to the reviewer’s suggestions, we have observed the surface morphology of the treated ACEC@FEP

using scanning electron microscope, and the image was added in Fig. S5 and S6. As can be seen that the surface of ACEC@FEP has little change after water droplet sliding and immersion in water, indicating its good stability. The relevant content in the manuscript has been revised and we hope to receive your approval. The details are as follows.

“As can be seen in Fig. 3d and S5, the surface morphology of ACEC@FEP hardly changed after the water droplets slid down continuously for 10,000 times (Fig. S5). The contact angle remained stable and the coating did not peel off after soaking in water for different time (Fig. S6).”

Supplementary Figure 5. a, b SEM images of ACEC@FEP and after the water droplets sliding down (10,000 times).

Supplementary Figure 6. a, b SEM images of ACEC@FEP and after immersion in water for 10 h.

4. For fog harvesting, a fog generator was used to simulate the fog environment. It's worth to noted that in real case, the fog in the air may be affected by the environment condition, especially humidity. The authors should better provide some data to verify the relationship of fog flows and humidity.

Response: We sincerely thank the reviewer's professional question and valuable suggestion. As the reviewer stated that the fog may be affected by the environment condition. Accordingly, we investigated the effect of fog flow rate on the relative humidity, and the related results were shown in Fig. S22 in the revised manuscript.

(1) A test chamber was designed to minimize the impact of the environment (such as wind). Additionally, a high-precision hygrometer was employed to record the relative humidity around the amphiphilic asymmetric surface.

Figure R 1 Optical photograph of the humidity test setup.

(2) In addition, the relative humidity at different fog flow was recorded and has been added in the revised manuscript (Fig. S22). As shown in Fig. S22, the relative humidity around the amphiphilic asymmetric surface could reach 99.9% successively as the fog flow increases from 100 mL/h to 300 mL/h. The manuscript has been revised according to the reviewer’s suggestions and the detailed description are as follows.

Supplementary Figure 22. As the fog flow rate increased from 100 mL/h to 300 mL/h, the relative humidity around the amphiphilic asymmetric surface could reach 99.9% successively (ambient temperature $\sim 25^{\circ}\text{C}$, ambient humidity $58 \pm 1\%$).

Details are as follows.

“In all of the above operations, the relative humidity around the amphiphilic asymmetric surface was maintained at 99.9% (Fig.S22).”

5. In the output performance of D-TENG, ion injection technique was applied to enhance the output voltage. Would it influence the surface wettability of FEP and how long it could be sustained? The authors may need to provide some data to prove the stability of triboelectric effect.

Response: We greatly appreciate the reviewer’s professional comments. According to the reviewer’s suggestion, we have investigated the surface wettability of FEP and stability of triboelectric effect. The details are as follows.

(1) The static contact angle and dynamic contact angle of FEP after ion injection were investigated by using a contact angle meter (Fig. R5). As shown in Fig. R5, the

contact angle value of the FEP and FEP after ion implantation remained consistent within 120 s. In addition, their corresponding advancing contact angle and receding contact angle were almost the same. These results indicate that ion injection has little effect on the wettability of the FEP.

Figure R 5. Static contact angle and dynamic contact angle of FEP and FEP after ion injection.

(2) Notably, it takes too much time to accumulate surface charge to saturation through continuous droplet impacting. Therefore, we injected negative ions (CO_3^- , NO_3^- , NO_2^- , O_3^- and O_2^-) generated by air ionization onto FEP, the surface charge rapidly increased to a higher level, even beyond saturation. Accordingly, the start-up time of the D-TENG could be effectively shortened, and the initial output performance could also be improved to some extent. In our manuscript, the ion injection is mainly used to shorten the time for D-TENG to reach stability. In addition, the transferred charge for ion injection and self-charging was recorded (Fig. S17 and S18). Compared with the droplet self-charging (at least 770 times), the transferred charge fell back and stabilized after nearly 300 droplets impacting after the ion injection (Fig. S18).

Supplementary Figure 17. As the droplets continue to impinge on the D-TENG, the amount of charge on the FEP surface increased gradually followed by remaining stable.

Supplementary Figure 18. After ion injection on the D-TENG surface, the transferred charge decreased rapidly from 371 nC followed by remaining stable.

6. For electrical measurement, approximately constant 96 μ L droplet impinged on the D-TENG and investigating the output performance by adjust different droplet release heights. Also, vary volume of droplet may and the impact angle may change the output. I would like to suggest the authors can add some statement about this.

Response: We greatly appreciate the reviewer’s professional suggestions and thank the reviewer again. Accordingly, the output performance was investigated by adjust the impact angle as well as the droplet volume, and the related results are shown in Fig. S19 and S20 in the revised manuscript.

(1) Fig. S19 shows the open-circuit voltage and transferred charge of the D-TENG at different impacting angles. As the impact angle increased from 15° to 75°, the open-circuit voltage increased from 80.1 V to 103.2 V, followed by a gradual decreased to 51.6 V, and the corresponding transferred charge increased from 27.6 nC to 42.4 nC and then decreased to 17.9 nC. The highest value was shown at 45°, which is consistent with the work of Wang et al. (Wang. et al. EcoMat 2021, 3:e12116, Xu. et al. Nature 2020, 7795:2020578).

Supplementary Figure 19. Open-circuit voltage and transfer charge of the D-TENG at different impacting angles. As the impact angle increased from 15° to 75°, the open-circuit voltage increased from 80.1 V to 103.2 V, followed by a gradual decreased to 51.6 V, and the corresponding transfer charge increased from 27.6 nC to 42.4 nC and then decreased to 17.9 nC. The highest value was shown at 45°, which is consistent

with the work of Wang et al. (Wang. et al. EcoMat 2021, 3:e12116, Xu. et al. Nature 2020, 7795:2020578).

(2) Fig. S20 shows the voltage and transferred charge of the D-TENG for different droplet volumes. As shown in Fig. S20, as the droplet volume increased from 54 μL to 96 μL , the open-circuit voltage increased from 61.2 V to 103.2 V, and the transferred charge increased from 22.1 nC to 42.4 nC. Therefore, the larger the droplet volume is, the better the electrical output performance is.

Supplementary Figure 20. Voltage and transferred charge of the D-TENG for different droplet volumes. As the droplet volume increased from 54 μL to 96 μL , the open-circuit voltage increased from 61.2 V to 103.2 V, the transferred charge increased from 22.1 nC to 42.4 nC.

Details are as follows.

“Subsequently, parameters such as impact angle, droplet volume and droplet release height were optimized (Fig. 5c, S19, and S20). Interestingly, Open-circuit voltage can reach 103.2 V, which could light 400 commercial LED bulbs.”

REVIEWER COMMENTS

Reviewer #1 (Remarks to the Author):

In the manuscript entitled "Bioinspired asymmetric amphiphilic surface for triboelectric enhanced efficient water harvesting," the author creatively uses the droplet triboelectric nanogenerators to provide the negative charge for electrostatically assisted fog collection. Moreover, a bioinspired asymmetric amphiphilic surface is applied to further improve system efficiency. After discussing with reviewers, the paper has sufficient data and clear expression. This water harvesting system design is valuable, but some comments need to be addressed before publication.

1. It is proved experimentally that the negative charge can enhance fog collection, which is attributed to strong Coulomb force from the applied charge. However, theoretical analysis is expected to be added to make this part more complete.
2. What is the relationship between bioinspired asymmetric amphiphilic surface and electrostatically assisted fog collection? Even if there is no bioinspired surface, D-TENG can also improve the fog collection ability of the system. It seems that bionic structure and electrostatic fog collection are two parallel tasks.
3. I think the core content of this article is the self-driven triboelectric adsorption system. The LED lamps are only used to prove open-circuit voltage and do not need to be mentioned in the abstract.
4. Figure 6e-f seems to illustrate that as the number of sliding droplets increases and time goes by, the capacitance becomes saturated, and the fog adsorption capacity gradually increases. The x-axis of these two figures should be consistent for the convenience of the reader's understanding.
5. In Figure 6g, "negative" is misspelled. And the text in the figure should be consistent with the description in the picture title.

Reviewer #2 (Remarks to the Author):

The authors have made good revision of their manuscript according to my comments. Therefore I am happy to suggest the acceptance of it.

Point-by-point Response to Reviewers' Comments (Manuscript ID: NCOMMS-22-10311A)

Reviewer: #1

In the manuscript entitled “Bioinspired asymmetric amphiphilic surface for triboelectric enhanced efficient water harvesting,” the author creatively uses the droplet triboelectric nanogenerators to provide the negative charge for electrostatically assisted fog collection. Moreover, a bioinspired asymmetric amphiphilic surface is applied to further improve system efficiency. After discussing with reviewers, the paper has sufficient data and clear expression. This water harvesting system design is valuable, but some comments need to be addressed before publication.

Response: We are deeply grateful for the review’s positive comments on our work as “*The paper has sufficient data and clear expression. This water harvesting system design is valuable*” and the valuable feedback that we have used to improve the quality of the manuscript. We strongly agree with the reviewer’s suggestions and thank the reviewer again.

1. It is proved experimentally that the negative charge can enhance fog collection, which is attributed to strong Coulomb force from the applied charge. However, theoretical analysis is expected to be added to make this part more complete.

Response: We greatly appreciate the reviewer’s professional comments, which are of great help to our work. According to the editor’s suggestions, the fog droplets in the single electrostatic field and the electric field applied charge were investigated in detail. The manuscript has been revised and the detailed description is as follows.

“It is assumed that the droplets do not collide with each other and that the electric field does not change as the droplets attach. As an electric field E generated after the FEP was charged, the motion state of the droplet in the electric field could be calculated by the formula 3⁸,

$$m \frac{du}{dt} = 6\pi\eta_g R_d (w - u) + qE \quad (3)$$

where m is the mass of the droplet, u is its velocity, R_d is its radius, q is its charge, t is the time, η_g is the air viscosity, and w is air velocity. The droplets were triboelectrically positive due to friction with the air, and the positive charges would be rearranged by polarization effects of the electrostatic field^{8,18}. Therefore, the droplet was mainly driven by the Coulomb force (to overcome the drag resistance F_{drag}), moved along the electric field lines towards the FEP, and finally attached to FEP surface (Fig. 6e). After the negative charge injection, the locally high charge density would lead to an inhomogeneous electric field. The droplets would move towards the FEP, and enrich in regions with high charge density (Fig. 6f). In addition, the agglomerated droplets on the FEP surface would also be driven toward this region⁴² (Fig. 6g).”

Fig. 6. e Schematic of droplets in electrostatic field. f, g Schematic of the droplets in the electric field applied charge.

2. What is the relationship between bioinspired asymmetric amphiphilic surface and electrostatically assisted fog collection? Even if there is no bioinspired surface, D-TENG can also improve the fog collection ability of the system. It seems that bionic structure and electrostatic fog collection are two parallel tasks.

Response: We greatly appreciate the reviewer’s valuable question, which really helps us to improve our work. In the water collection system, the biomimetic asymmetric amphiphilic surface dominates and runs through the whole collection process, while the electrostatically assisted part accelerates the merging and detachment of droplets (t_2+t_3) through local coalescence and sliding. Both of them complement each other and are indispensable. Specifically, the hydrophilic component and the spines structure provide sites for the attachment of the fog, and then the condensed droplets creep to the roots of the spines due to the Laplace pressure. When the additional pressure from gravity and the Laplace pressure are greater than the energy barrier at the junction, the droplet enters the hydrophobic channel. Further, electrostatic assistance accelerates droplet shedding by reducing the droplets coalescence spacing and increasing the local droplet density on the hydrophobic channel surface. It is worth noting that the static electricity generated from the contact electrification between the water droplet and the hydrophobic channel as well as the D-TENG. If the electrostatic effect is used to collect fog alone, it is inevitable to introduce external energy to provide a high voltage of tens of thousands of volts (Damak M, et al. *Science Advances*, 2018, 4(6): eaao5323). To reduce energy consumption, self-power electrostatic interactions were employed to enhance water harvesting.

3. I think the core content of this article is the self-driven triboelectric adsorption system. The LED lamps are only used to prove open-circuit voltage and do not need to be mentioned in the abstract.

Response: We apologize for using imprecise languages in the manuscript. According to the reviewer’s suggestions, the manuscript has been revised and the detailed description is as follows.

“the droplet triboelectric nanogenerator (D-TENG) converts the mechanical energy generated by droplets falling into electrical energy through the volume effect, achieving an excellent output performance, and further enhancing the electrostatic adsorption by means of external charges”

4. Figure 6e-f seems to illustrate that as the number of sliding droplets increases

and time goes by, the capacitance becomes saturated, and the fog adsorption capacity gradually increases. The x-axis of these two figures should be consistent for the convenience of the reader's understanding.

Response: We highly appreciate the reviewer's professional comments, which kindly help us to further improve our work. According to the reviewer's suggestion, we have revised in the manuscript. The details are as follows.

Fig. 6. h Charge transfer of a thin FEP strip (5 x 50 mm) as water droplets slide off.

5. In Figure 6g, “negative” is misspelled. And the text in the figure should be consistent with the description in the picture title.

Response: We apologize for the writing mistake and highly appreciate the reviewers for their careful review. We have carefully checked the full manuscript. Afterwards, the Fig. 6g has been revised (Fig. 6j) and we hope to receive your approval. Details are as follows.

Fig. 6. j Water harvesting rate under grounding, self-triboelectrification, and external charge.

Reviewer: #2

The authors have made good revision of their manuscript according to my comments. Therefore, I am happy to suggest the acceptance of it.

Response: Thanks very much for your kind work and consideration on publication of

our paper. On behalf of my co-authors, we would like to express our great appreciation to you.

REVIEWERS' COMMENTS

Reviewer #1 (Remarks to the Author):

The authors have addressed all the comments and revised the manuscript accordingly. Therefore, I recommend publication.